# LTU Attacker for Membership Inference

**Joseph Pedersen** [1,*] , **Rafael Muñoz-Gómez** [2] , **Jiangnan Huang** [2], **Haozhe Sun** [2], **Wei-Wei Tu** [3,4] **and Isabelle Guyon** [2,4]

1  Department of Industrial and Systems Engineering, Rensselaer Polytechnic Institute, Troy, NY 12180, USA
2  LISN/CNRS/INRIA, Paris-Saclay University, 3 Rue Joliot Curie Bâtiment Breguet, 91190 Gif-sur-Yvette, France; rafaelmugo07@gmail.com (R.M.-G.); jiangnan2217@gmail.com (J.H.); sunhaozhe275940200@gmail.com (H.S.); guyon@chalearn.org (I.G.)
3  4Paradigm, 66 Qinghe Middle Street, Beijing 100089, China; tuww.cn@gmail.com
4  ChaLearn, 397 Schimke Road, Alpine County, CA 95223, USA
*  Correspondence: joseph.m.pedersen@gmail.com

**Abstract:** We address the problem of defending predictive models, such as machine learning classifiers (Defender models), against membership inference attacks, in both the black-box and white-box setting, when the trainer and the trained model are publicly released. The Defender aims at optimizing a dual objective: utility and privacy. Privacy is evaluated with the membership prediction error of a so-called "Leave-Two-Unlabeled" LTU Attacker, having access to all of the Defender and Reserved data, except for the membership label of one sample from each, giving the strongest possible attack scenario. We prove that, under certain conditions, even a "naïve" LTU Attacker can achieve lower bounds on privacy loss with simple attack strategies, leading to concrete necessary conditions to protect privacy, including: preventing over-fitting and adding some amount of randomness. This attack is straightforward to implement against any model trainer, and we demonstrate its performance against MemGaurd. However, we also show that such a naïve LTU Attacker can fail to attack the privacy of models known to be vulnerable in the literature, demonstrating that knowledge must be complemented with strong attack strategies to turn the LTU Attacker into a powerful means of evaluating privacy. The LTU Attacker can incorporate any existing attack strategy to compute individual privacy scores for each training sample. Our experiments on the QMNIST, CIFAR-10, and Location-30 datasets validate our theoretical results and confirm the roles of over-fitting prevention and randomness in the algorithms to protect against privacy attacks.

**Keywords:** machine learning; privacy attacks; membership inference

## 1. Introduction

Large companies are increasingly reluctant to let any information out, for fear of privacy attacks and possible ensuing lawsuits. Even government agencies and academic institutions, whose charter is to disseminate data and results publicly, must be careful. Hence, we are in great need of simple and provably effective protocols to protect data, while ensuring that some utility can be derived from them. Though critical sensitive data must never leave the source organization (Source)—company, government, or academia—an authorized researcher (Defender) may gain access to them within a secured environment to analyze them and produce models (Product). The Source may desire to release the Product, provided that desired levels of utility and privacy are met. We consider the most complete release of model information, including the Defender trainer, with all its settings, and the trained model. This enables "white-box attacks" from potential attackers [1]. We devise an evaluation apparatus to help the Source in its decision whether or not to release the Product (Figure 1). The setting considered is that of "membership inference attack", in which an attacker seeks to uncover whether given samples, distributed similarly as the Defender training dataset, belong or not to such dataset [2]. The apparatus includes an Evaluator and an LTU Attacker. The Evaluator performs a hold-out leave-two-unlabeled (LTU) evaluation,

giving the LTU Attacker access to extensive information: all the Defender and Reserved data, except for the membership label of one sample from each. The contributions of our paper include this new evaluation apparatus. Its soundness is backed by some initial theoretical analyses and by preliminary experimental results. These indicate that even naïve attacks can defeat privacy protections with this framework, and yet some Defender models can protect data privacy while retaining utility in such extreme attack conditions.

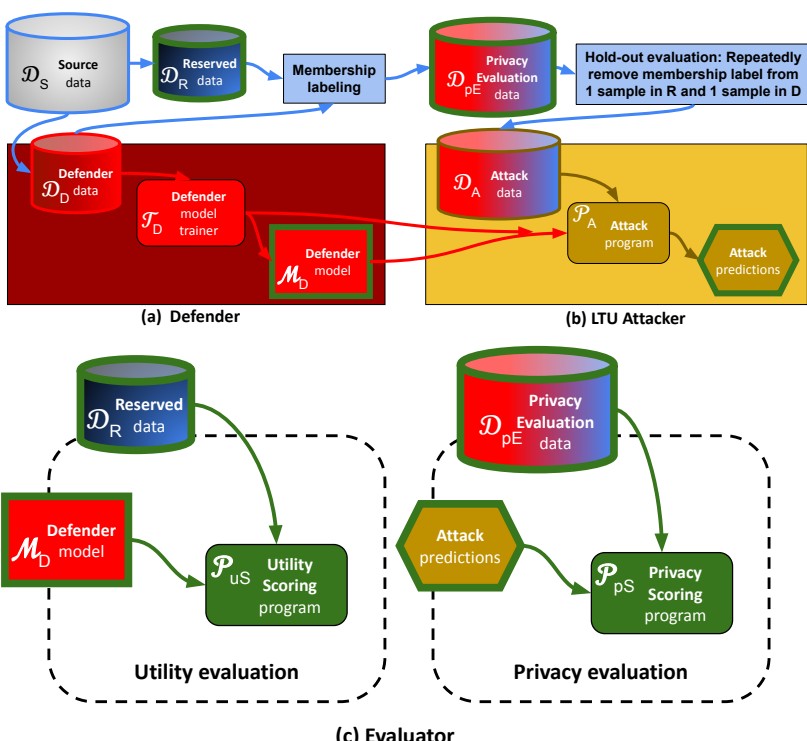

**Figure 1.** Methodology flow chart. (**a**) Defender: Source data are divided into Defender data, to train the model under attack (Defender model) and Reserved data to evaluate such a model. The Defender model trainer creates a model optimizing a utility objective, while being as resilient as possible to attacks. (**b**) LTU Attacker: The evaluation apparatus includes an LTU Attacker and an Evaluator: The evaluation apparatus performs a hold-out evaluation leaving two unlabeled examples (LTU) by repeatedly providing the LTU Attacker with ALL of the Defender and Reserved data samples, together with their *membership origin*, hiding only the membership label of 2 samples. The LTU Attacker must turn in the membership label (Defender data or Reserved data) of these 2 samples (Attack predictions). (**c**) Evaluator: The Evaluator computes two scores: LTU Attacker prediction error (Privacy metric), and Defender model classification performance (Utility metric).

## 2. Related Work

Membership inference attacks (MIA) have been extensively studied in recent years. Ref. [3] developed a privacy framework called "Membership Privacy", establishing a family of related privacy definitions. Ref. [2] explored the first MIA scenario, in which an attacker has black-box query access to a classification model $f$ and can obtain the prediction vector of the data record $x$ given as input. Ref. [4] proposed a metric inspired from Differential Privacy to measure the privacy risk of each training record, based on the impact it has on the learning algorithm. Ref. [5] derive a tighter bound on the membership inference attack accuracy against models that are differentially private. Similarly, Ref. [6] incorporates a fine-grained analysis on the systematic evaluation of privacy risk. The Bayesian metric proposed is defined as the posterior probability that a given input sample is from the training set after observing the target model's behavior over that sample. Ref. [7] explores a more realistic scenario. They consider skewed priors where only a small fraction of the samples belong to the training set, and its attack strategy is focused on selecting the best

inference thresholds. In contrast, our LTU Attacker is focused on a worst-case analysis, rather than on the most practical scenarios.

Ref. [8] studied the connection between overfitting and membership inference, showing that overfitting is a sufficient condition to guarantee success of the adversary. Ref. [9] continued exploring MIAs in the black-box model setting, considering different scenarios according to the prior knowledge that the adversary has about the training data: black-box, grey-box, and white-box. Recent work also addressed membership inference attacks against generative models [10–12]. This paper focuses on the attack of discriminative models in an all 'knowledgeable scenario', both from the point of view of model and data.

Several frameworks have been proposed to mitigate attacks, among which Differential Privacy [13] has become a reference method. Work in [14,15] shows how to implement this technique in deep learning. Using DP to protect against attacks comes at the cost of decreasing the model's utility. Regularization approaches have been investigated in an effort to increase model robustness against privacy attacks while retaining most utility. One of them inspired our idea to defend against attacks in an adversarial manner: Domain-adversarial training [16] introduced in the context of domain adaptation. Ref. [17] will later use this technique to defend against MIA. Ref. [18] helped bridge the gap between membership inference and domain adaptation.

Most literature addressing MIA considers a black-box scenario, where the adversary only has access to the model through an API and very little knowledge about the training data. Closest to the scenario considered in this paper, the work of [1] analyzes attackers having all information about the neural network under attack, including inner layer outputs, allowing them to exploit privacy vulnerabilities of the SGD algorithm. However, contrary to the LTU Attacker we are introducing, the authors' adversary executes the attack in an unsupervised way, without having access to membership labels of any data sample. Additionally, the work of [19] consider a similar scenario of giving an attacker two datasets differing by one sample, but focus their work on DP-SGD, and give the attacker access to intermediate computations performed during training. We assume that the attacker had no access to the trainer or model during training. Bayes optimal strategies have been examined in [20]; showing that, under some assumptions, the optimal inference depends only on the loss. Recent work in [21] also aims to design the best possible adversary, defined in terms of the Bayes Optimal Classifier, to estimate privacy leakage of a model.

## 3. Problem Statement and Methodology

We consider the scenario in which an owner of a data Source $\mathcal{D}_S$ wants to create a predictive model trained on some of those data, but needs to ensure that privacy is preserved. In particular, we focus on privacy from membership inference. The data owner entrusts an agent called Defender with creating such a model, giving him access to a random sample $\mathcal{D}_D \subset \mathcal{D}_S$ (Defender dataset). We denote by $\mathcal{M}_D$ the trained model (Defender model) and by $\mathcal{T}_D$ the algorithm used to train it (Defender trainer). The data owner wishes to release $\mathcal{M}_D$, and eventually $\mathcal{T}_D$, provided that certain standards of privacy and utility of $\mathcal{M}_D$ and $\mathcal{T}_D$ are met. To evaluate such utility and privacy, the data owner reserves a dataset $\mathcal{D}_R \subset \mathcal{D}_S$, disjointed from $\mathcal{D}_D$, and gives both $\mathcal{D}_D$ and $\mathcal{D}_R$ to a trustworthy Evaluator agent. The Evaluator tags the samples with dataset "membership labels": *Defender* or *Reserved*. Then, the Evaluator performs repeated rounds, consisting in randomly selecting one Defender sample $d$ and one Reserved sample $r$, and giving to an LTU Attacker an almost perfect attack dataset $\mathcal{D}_A = \mathcal{D}_D - \{\texttt{membership}(d)\} \cup \mathcal{D}_R - \{\texttt{membership}(r)\}$, removing only the membership labels of the two selected samples. The two unlabeled samples are referred to as $u_1$ and $u_2$, with each being equally likely to be from the Defender dataset. We refer to this procedure as "Leave Two Unlabeled" (LTU), see Figure 1. The LTU Attacker also has access to the Defender trainer $\mathcal{T}_D$ (with all its hyper-parameter settings), and the trained Defender model $\mathcal{M}_D$. This is the worst-case scenario in terms of attacker knowledge/access, because the only other information that can be used to attack the membership, would be to already have the membership labels. The attacker is tasked

with correctly predicting which of the two samples $d$ and $r$ belongs to $\mathcal{D}_D$ (independently for each LTU round, forgetting everything at the end of a round).

We use the *LTU membership classification accuracy $A_{ltu}$* from $N$ independent LTU rounds (as defined above), to define a global privacy score as:

$$\texttt{Privacy} = \min\{2\,(1 - A_{ltu}), 1\}$$
$$\pm\, 2\sqrt{A_{ltu}(1 - A_{ltu})/N}\,, \tag{1}$$

where the error bar is an estimator of the standard error of the mean (approximating the Binomial law with the Normal law, see, e.g., [22]). The weaker the performance of the LTU Attacker ($A_{ltu} \simeq 0.5$ for random guessing), the larger $\texttt{Privacy}$, and the better $\mathcal{M}_D$ should be protected from attacks. We can also determine an **individual membership inference privacy score** for any sample $d \in \mathcal{D}_D$ by using that sample for all $N$ rounds, and only drawing $r \sim \mathcal{D}_R$ at random (Similarly, we can determine an individual non-membership inference privacy score for any sample $r \in \mathcal{D}_R$ by using that sample for all $N$ rounds, and only drawing $d$ at random). See the example in Appendix C.

The Evaluator also uses $\mathcal{D}_{uE} = \mathcal{D}_R$ to evaluate the **utility of the Defender model $\mathcal{M}_D$**. We focus on multi-class classification for $c$ classes, and measure utility with the *classification accuracy $A_D$* of $\mathcal{M}_D$, defining utility as:

$$\texttt{Utility} = \max\{(c\,A_D - 1)/(c - 1), 0\}$$
$$\pm\, c\sqrt{A_D(1 - A_D)/|\mathcal{D}_R|}\,, \tag{2}$$

Although the LTU Attacker is all knowledgeable, we still need to endow it with an algorithm to make membership predictions. In Figure 2 we propose a taxonomy of LTU Attackers. In each LTU round, let $u_1$ and $u_2$ be the samples that were deprived of their labels. The taxonomy has 2 branches:

- Attack on $\mathcal{M}_D$ alone: (1) Simply use a generalization **Gap-attacker**, which classifies $u_1$ as belonging to $\mathcal{D}_D$ if the loss function of $\mathcal{M}_D(u_1)$ is smaller than that of $\mathcal{M}_D(u_2)$ (works well if $\mathcal{M}_D$ overfits $\mathcal{D}_D$); (2) Train a $\mathcal{M}_D$-**attacker** $\mathcal{M}_A$ to predict membership, using as input any internal state or the output of $\mathcal{M}_D$, and using $\mathcal{D}_A$ as training data. Then use $\mathcal{M}_A$ to predict the labels of $u_1$ and $u_2$.
- Attack on $\mathcal{M}_D$ and $\mathcal{T}_D$: Depending on whether the Defender trainer $\mathcal{T}_D$ is a white-box from which gradients can be computed, define $\mathcal{M}_A$ by: (3) Training two mock Defender models $\mathcal{M}_1$ and $\mathcal{M}_2$, one using $(\mathcal{D}_D - \{d\}) \cup \{u_1\}$ and the other using $(\mathcal{D}_D - \{d\}) \cup \{u_2\}$, with the trainer $\mathcal{T}_D$. If $\mathcal{T}_D$ is deterministic and independent of sample ordering, either $\mathcal{M}_1$ or $\mathcal{M}_2$ should be identical to $\mathcal{M}_D$, and otherwise one of them should be "closer" to $\mathcal{M}_D$. The sample corresponding to the model closest to $\mathcal{M}_D$ is classified as being a member of $\mathcal{D}_D$. (4) Performing one gradient learning step with either $u_1$ or $u_2$ using $\mathcal{T}_D$, starting from the trained model $\mathcal{M}_D$, and compare the gradient norms.

A variety of Defender strategies might be considered:

- Applying over-fitting prevention (regularization) to $\mathcal{T}_D$;
- Applying Differential Privacy algorithms to $\mathcal{T}_D$;
- Training $\mathcal{T}_D$ in a semi-supervised way (with transfer learning) or using synthetic data (generated with a simulator trained with a subset of $\mathcal{D}_D$);
- Modifying $\mathcal{T}_D$ to optimize both utility and privacy.

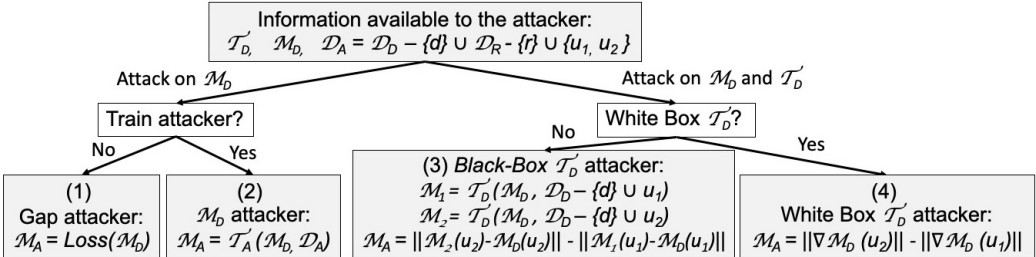

**Figure 2.** Taxonomy of LTU Attacker. **Top:** Any LTU Attacker has available the Defender trainer $\mathcal{T}_D$, the trained Defender model $\mathcal{M}_D$, and attack data $\mathcal{D}_A$ including (almost) all the Defender data $\mathcal{D}_D$ and Reserved data $\mathcal{D}_R$ $\mathcal{D}_A = \mathcal{D}_D - \{\texttt{membership}(d)\} \cup \mathcal{D}_R - \{\texttt{membership}(r)\}$. However, it may use only part of this available knowledge to conduct attacks. $r$ and $d$ are two labeled examples belonging $\mathcal{D}_R$ and $\mathcal{D}_D$, respectively, and $u_1$ and $u_2$ are two unlabeled examples, one from $\mathcal{D}_R$ and one from $\mathcal{D}_D$ (ordered randomly). **Bottom Left:** Attacker $\mathcal{M}_A$ targets only the trained Defender model $\mathcal{M}_D$. **Bottom Right:** $\mathcal{M}_A$ targets both $\mathcal{M}_D$ and its trainer $\mathcal{T}_D$.

## 4. Theoretical Analysis of Naïve Attackers

We present several theorems outlining weaknesses of the Defender that are particularly easy to exploit by a black-box LTU Attacker, not requiring training a sophisticated attack model $\mathcal{M}_A$ (we refer to such attackers as "naïve"). First, we prove, in the context of the LTU procedure, theorems related to an already known result connecting privacy and over-fitting: Defender trainers that overfit the Defender data lend themselves to easy attacks [8]. The attacker can simply exploit the loss function of the Defender (which should be larger on Reserved data than on Defender data). The last theorem concerns deterministic trainers $\mathcal{T}_D$: We show that the LTU Attacker can defeat them with 100% accuracy, under mild assumptions. Thus, Defenders must introduce some randomness in their training algorithm to be robust against such attacks [23].

Throughout this analysis, we use the fact that our LTU methodology simplifies the work for the LTU Attacker since it is always presented with pairs of samples for which exactly one is in the Defender data. This can give it very simple attack strategies. For example, for any real valued function $f(x)$, with $x \in \mathcal{D}_S$, let $r$ be drawn uniformly from $\mathcal{D}_R$ and $d$ be drawn uniformly from $\mathcal{D}_D$, and define:

$$p_R = \Pr_{\substack{u_1 \sim \mathcal{D}_R \\ u_2 \sim \mathcal{D}_D}} [f(u_1) > f(u_2)] \tag{3}$$

$$p_D = \Pr_{\substack{u_1 \sim \mathcal{D}_R \\ u_2 \sim \mathcal{D}_D}} [f(u_1) < f(u_2)] \tag{4}$$

Thus, $p_R$ is the probability that discriminant function $f$ "favors" Reserved data while $p_D$ is the probability with which it favors the Defender data. $p_R > p_D$ occurs if for a larger number of random pairs $f(x)$ is larger for Reserved data than for Defender data. If the probability of a tie is zero, then $p_R + p_D = 1$.

**Theorem 1.** *If there is any function $f$ for which $p_R > p_D$, an LTU Attacker exploiting that function can achieve an accuracy $A_{ltu} \geq \frac{1}{2} + \frac{1}{2}(p_R - p_D)$.*

**Proof.** A simple attack strategy would be predict that the unlabeled sample with the smaller value of $f(x)$ belongs to the *Defender* data, with ties (i.e., when $f(u_1) = f(u_2)$) decided by tossing a fair coin. This strategy would give a correct prediction when $f(r) > f(d)$, which occurs with probability $p_R$, and would be correct half of the time when $f(r) = f(d)$, which occurs with probability $(1 - (p_r + p_d))$. This gives a classification accuracy:

$$A_{ltu} = p_R + \frac{1}{2}(1 - p_R - p_D) = \frac{1}{2} + \frac{1}{2}(p_R - p_D) . \tag{5}$$

$\square$

This is similar to the threshold adversary of [8], except that the LTU Attacker does not need to know the exact conditional distributions, since it can discriminate pairwise. The most obvious candidate function $f$ is the loss function used to train $\mathcal{M}_D$ (we call this a naïve attacker), but the LTU Attacker can mine $\mathcal{D}_A$ to potentially find more discriminative functions, or multiple functions to bag, and use $\mathcal{D}_A$ to compute very good estimates for $p_R$ and $p_D$. We verify that an LTU Attacker using an $f$ function making perfect membership predictions (e.g., having the knowledge of the *entire* Defender dataset and using the nearest neighbor method) would get $A_{ltu} = 1$, if there are no ties. Indeed, in that case, $p_R = 1$ and $p_D = 0$.

In our second theorem, we show that the LTU Attacker can attain an analogous lower bound on accuracy connected to overfitting as the bounded loss function (BLF) adversary of [8].

**Theorem 2.** *If the loss function $\ell(x)$ used to train the Defender model is bounded for all $x$, without loss of generality $0 \leq \ell(x) \leq 1$ (since loss functions can always be re-scaled), and if $e_R$, the expected value of the loss function on the Reserved data, is larger than $e_D$, the expected value of the loss function on the Defender data, then a lower bound on the accuracy of the LTU Attacker is given by the following function of the generalization error gap $e_R - e_D$:*

$$A_{ltu} \geq \frac{1}{2} + \frac{1}{2}(e_R - e_D) \tag{6}$$

**Proof.** If the order of the pair $(u_1, u_2)$ is random and the loss function $\ell(x)$ is bounded by $0 \leq \ell(x) \leq 1$, then the LTU Attacker could predict $u_1 \in \mathcal{D}_R$ with probability $\ell(u_1)$, by drawing $z \sim U(0,1)$ and predicting $u_1 \in \mathcal{D}_R$ if $z < \ell(u_1)$, and $u_1 \in \mathcal{D}_D$ otherwise. This gives the desired lower bound, derived in more detail in Appendix A:

$$
\begin{aligned}
A_{ltu} &= \frac{1}{2} \Pr_{u_1 \sim \mathcal{D}_R}[z < \ell(u_1)] + \frac{1}{2}\left(1 - \Pr_{u_1 \sim \mathcal{D}_D}[z < \ell(u_1)]\right) \\
&= \frac{1}{2} \mathop{\mathbb{E}}_{u_1 \sim \mathcal{D}_R}[\ell(u_1)] + \frac{1}{2} - \frac{1}{2} \mathop{\mathbb{E}}_{u_1 \sim \mathcal{D}_D}[\ell(u_1)] \\
&= \frac{1}{2} + \frac{e_R - e_D}{2} \\
&\text{where } e_R := \mathop{\mathbb{E}}_{u_1 \sim \mathcal{D}_R}[\ell(u_1)] \text{ and } e_D := \mathop{\mathbb{E}}_{u_1 \sim \mathcal{D}_D}[\ell(u_1)]
\end{aligned}
\tag{7}
$$

$\square$

This is only a lower bound on the accuracy of the attacker, connected to the main difficulty in machine learning-overfitting of the loss function. Other attack strategies may be more accurate. However, neither of the attack strategies in Theorems 1 and 2 is dominant over the other, as shown in Appendix B. The strategy in Theorem 1 is more widely applicable, since it does not require the function to be bounded.

In the special case when the loss function used to train the Defender model is the 0–1 loss, and that is used to attack (i.e., $f = \ell$), the strategies in Theorems 1 and 2 are different, but have the same accuracy:

$$
\begin{aligned}
p_R &= \Pr_{u \sim \mathcal{D}_R}[\ell(u) = 1]\left(1 - \Pr_{u \sim \mathcal{D}_D}[\ell(u) = 1]\right) \\
p_D &= \left(1 - \Pr_{u \sim \mathcal{D}_R}[\ell(u) = 1]\right) \Pr_{u \sim \mathcal{D}_D}[\ell(u) = 1] \\
p_R - p_D &= \Pr_{u_1 \sim \mathcal{D}_R}[\ell(u) = 1] - \Pr_{u \sim \mathcal{D}_D}[\ell(u) = 1] \\
&= e_R - e_D
\end{aligned}
$$

Note that $u$ is a dummy variable. The first line of the derivation is due to the fact that the only way the loss on the Reserved set can be greater than the loss on the Defender set is

if the loss on the Reserved set is 1, which has probability $Pr_{u \sim \mathcal{D}_R}[\ell(u) = 1]$, and the loss on the Defender set is zero, which has probability $1 - Pr_{u \sim \mathcal{D}_D}[\ell(u) = 1]$. The second line is derived similarly.

**Theorem 3.** *If the Defender trainer $\mathcal{T}_D$ is deterministic, invariant to the order of the training data, and injective, then the LTU Attacker has an optimal attack strategy, which achieves perfect accuracy.*

**Proof.** The proof uses the fact that the LTU Attacker knows all of the Defender dataset except one sample, and knows that the missing sample is either $u_1$ or $u_2$. Therefore, the attack strategy is to create two models, one trained on $u_1$ combined with the rest of the Defender dataset, and the other trained on $u_2$ combined with the rest of the Defender dataset. Since the Defender trainer is deterministic, one of those two models will match the Defender model, revealing which unlabeled sample belonged in the Defender dataset.

Formally, denote the subset of $\mathcal{D}_A$ labeled "Defender" as $\mathcal{D}_D - \{d\}$, and the two membership unlabeled samples as $u_1$ and $u_2$. The attacker can use the Defender trainer with the same hyper-parameters on $(\mathcal{D}_D - \{d\}) \cup \{u_1\}$ to produce model $\mathcal{M}_1$ and on $(\mathcal{D}_D - \{d\}) \cup \{u_2\}$ to produce model $\mathcal{M}_2$.

By definition of the LTU Attacker, the missing sample $d$ is either $u_1$ or $u_2$, and $\mathcal{D}_D \cap \mathcal{D}_R = \varnothing$, so $u_1 \neq u_2$. There are two possible cases. If $u_1 = d$, then $\mathcal{D}_D = (\mathcal{D}_D - \{d\}) \cup \{u_1\}$, so that $\mathcal{M}_1 = \mathcal{M}_D$, since $\mathcal{T}_D$ is deterministic and invariant to the order of the training data. However, $\mathcal{D}_D \neq (\mathcal{D}_D - \{d\}) \cup \{u_2\}$, since $u_2 \neq u_1$, so $\mathcal{M}_2 \neq \mathcal{M}_D$, since $\mathcal{T}_D$ is also injective. Therefore, the LTU Attacker can know, with no uncertainty, that $u_1$ has membership label "Defender" and $u_2$ has membership label "Reserved". The other case, for $u_2 = d$, has a symmetric argument. □

Under the hypotheses above, the LTU Attacker achieves the optimal Bayesian classifier using:

$$Pr[u_i \in \mathcal{D}_D | \mathcal{M}_i = \mathcal{M}_D] = 1$$
$$Pr[u_i \in \mathcal{D}_R | \mathcal{M}_i = \mathcal{M}_D] = 0$$
$$Pr[u_i \in \mathcal{D}_D | \mathcal{M}_i \neq \mathcal{M}_D] = 0$$
$$Pr[u_i \in \mathcal{D}_R | \mathcal{M}_i \neq \mathcal{M}_D] = 1$$

## 5. Data and Experimental Setting

We are using three datasets in our experiments: CIFAR-10 [24], QMNIST [25], and Location-30 [2]. CIFAR-10 is an object classification dataset with 10 different classes, well-known as a benchmark for membership inference attacks [2,11,26]. QMNIST [25] is a hand-written digit recognition dataset, similarly preprocessed as the well-known MNIST [27], but including the whole original NIST Special Database 19 (https://www.nist.gov/srd/nist-special-database-19 accessed on 4 May 2022) data (402,953 images). QMNIST includes meta-data that MNIST was deprived of, including writer IDs and its origin (high-school students or Census Bureau employees), which could be used in future studies of attribute or property inference attack. They are not used in this work. Location-30 [2] was created from "check-ins" on the Foursquare social network. It is a sample of 5010 users with 446 binary features, grouped into 30 clusters (labeled by cluster).

To speed up our experiments, we preprocessed the data using a backbone neural network pretrained on some other dataset, and used the representation of the second last layer of the network. For QMNIST we used VGG19 [28] pretrained on Imagenet [29]. For CIFAR-10, we rely on Efficient-netv2 [30] pretrained on Imagenet21k and finetuned on CIFAR-100.

The data were then split as follows: The 402,953 QMNIST images were shuffled, then separated into 200,000 samples for Defender data and 202,953 for Reserved data. The CIFAR-10 data were also shuffled and split evenly (30,000/30,000 approximately).

## 6. Results

*6.1. Black-Box Attacker*

For our first set of experiments, we trained and evaluated various algorithms of the scikit-learn library accessed on 4 May 2022 as Defender model and evaluated the utility and privacy, based on two subsets of data: (1) 1600 random examples from the Defender (training) data and (2) 1600 random examples from the Reserved data (used to evaluate utility and privacy). We performed $N = 100$ independent LTU rounds and then computed Privacy based on the LTU membership classification accuracy through Equation (1). The utility of the model was obtained with Equation (2). We used a Black-box LTU Attacker (number (3) in Figure 2). The results shown in Table 1 are averaged over 3 trials (Code is available at https://github.com/JiangnanH/ppml-workshop/blob/master/generate_table_v1.py accessed on 4 May 2022). The first few lines (gray shaded) are deterministic methods (whose trainer yields to the same model regardless of random seeds and sample order). For these lines, consistent with Theorem 3, Privacy is zero, in all columns (Results may vary depending upon which scikit-learn output method is used (`predict_proba()`, `decision_function()`, `density_function()`), or `predict()`. To achieve zero Privacy, consistent with the theory, the method `predict()` should be avoided). The algorithms use default scikit-learn hyper-parameter values. In the first two result columns, the Defender trainers are forced to be deterministic by seeding all random number generators. In the first column, the sample order is fixed to the order used by the Defender trainer, while in the second one it is not. Privacy in the first column is near zero, consistent with the theory. In the second column, this is also verified for methods independent of sample order. The third result column corresponds to varying the random seed, hence algorithms including some level of randomness have an increased level of privacy.

The results of Table 1 show that there is no difference between the column 2 and 3; suggesting that just the randomness associated to altering the order of the training samples is enough to make the strategy fail. These results also expose one limitation of black-box attacks: example-based methods (e.g., SVC), which store examples in the model, obviously violate privacy. However, this is not detected by a black-box LTU Attacker, if they are properly regularized and/or involve some degree or randomness. White-box attackers solve this problem.

In our next set of experiments, we attack a neural network classifier defended with MemGuard [31]. We use again use "naïve" white-box attack (method (3) in Figure 2). This time our attack program is to use the Defender model trainer to train many candidate models on random splits of half of the data, after which IN predictions are made on the half used for training and OUT predictions are made on the other half. These are used to create distributions for each sample of the predicted logits of the true class when the sample was IN or OUT of the training set. From these, we can use the Bayes theorem to attack the predictions of the Defender model. We find that even this straightforward black-box attack in the LTU framework can predict the membership of some samples with high confidence, as can be seen in Figure 3. The AUROC for the membership predictions was 0.8, and the TPR was much higher than the FPR in the low FPR regime as seen in Figure 4, indicating samples whose membership can be inferred with very low error.

**Table 1.** Utility and privacy of QMNIST and CIFAR-10 of different scikit-learn models with three levels of randomness: original sample order + fixed random seed (no randomness); random sample order + fixed random seed; random sample order + random seed. The Defender data and Reserved data have both 1600 examples. All numbers shown in the table have *at least* two significant digits (standard error lower than 0.004). For model implementations, we use scikit-learn (version 0.24.2) with default values. Shaded in gray: fully deterministic models with Privacy $\equiv 0$.

| **QMNIST** Utility \| Privacy | Orig. order + Seeded | Rand. order + Seeded | Not Seeded |
|---|---|---|---|
| Logistic lbfgs | 0.92\|0.00 | 0.91\|0.00 | 0.91\|0.00 |
| Bayesian ridge | 0.92\|0.00 | 0.92\|0.00 | 0.89\|0.00 |
| Naive Bayes | 0.70\|0.00 | 0.70\|0.00 | 0.70\|0.00 |
| SVC | 0.91\|0.00 | 0.91\|0.00 | 0.88\|0.00 |
| KNN* | 0.86\|0.27 | 0.86\|0.27 | 0.83\|0.18 |
| LinearSVC | 0.92\|0.00 | 0.92\|0.69 | 0.91\|0.63 |
| SGD SVC | 0.90\|0.03 | 0.92\|1.00 | 0.89\|1.00 |
| MLP | 0.90\|0.00 | 0.90\|0.97 | 0.88\|0.93 |
| Perceptron | 0.90\|0.04 | 0.91\|1.00 | 0.92\|1.00 |
| Random Forest | 0.88\|0.00 | 0.88\|0.99 | 0.85\|1.00 |

| **CIFAR-10** Utility \| Privacy | Orig. order + Seeded | Rand. order + Seeded | Not Seeded |
|---|---|---|---|
| Logistic lbfgs | 0.95\|0.00 | 0.95\|0.00 | 0.95\|0.00 |
| Bayesian ridge | 0.91\|0.00 | 0.90\|0.00 | 0.90\|0.00 |
| Naive Bayes | 0.89\|0.00 | 0.89\|0.01 | 0.89\|0.00 |
| SVC | 0.95\|0.00 | 0.94\|0.00 | 0.95\|0.00 |
| KNN* | 0.92\|0.44 | 0.91\|0.49 | 0.92\|0.49 |
| LinearSVC | 0.95\|0.00 | 0.95\|0.26 | 0.95\|0.22 |
| SGD SVC | 0.94\|0.32 | 0.94\|0.98 | 0.93\|0.99 |
| MLP | 0.95\|0.00 | 0.94\|0.98 | 0.95\|0.97 |
| Perceptron | 0.94\|0.26 | 0.94\|1.00 | 0.93\|0.96 |
| Random Forest | 0.92\|0.00 | 0.93\|0.99 | 0.91\|0.92 |

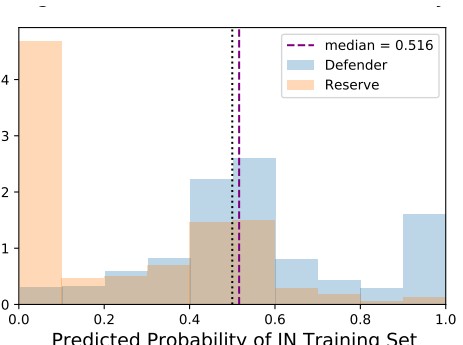
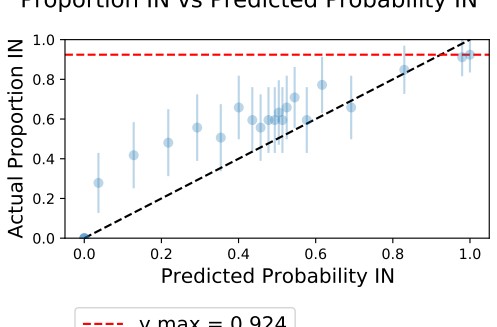

**Figure 3. Left:** Histograms of predicted probabilities of membership IN the Defender set, for the Defender set and the Reserve set, each of size 1000. **Right:** Scatter plot comparing the actual proportions of being IN the Defender set with the predicted probabilities of membership IN the Defender set, for groups of 40 samples with similar predictions (with error bars for proportion). Many Defender samples are correctly predicted members with high confidence.

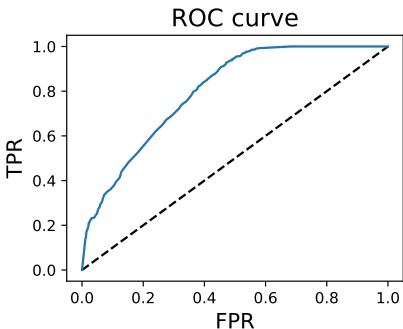

**Figure 4.** The ROC curve for the membership predictions from a model trained on Location-30 and defended by MemGuard. The AUROC was 0.8, and the TPR was much higher than the FPR in the low FPR regime.

### 6.2. White-Box Attacker

We implemented a white-box attacker based on gradient calculations (method (4) in Figure 2). We evaluated the effect of the proposed attack with QMNIST on two types of Defender models: a deep neural network (DNN) trained with supervised learning or with unsupervised domain adaptation (UDA) [32] (Code is available at https://github.com/JiangnanH/ppml-workshop#white-box-attacker accessed on 4 May 2022).

For supervised learning, we used ResNet50 [33] as the backbone neural network, which is pre-trained on ImageNet [29]. We then retrained all its layers on the Defender set of QMNIST. The results are reported in Table 2, line "Supervised". With the very large Defender dataset we are using for training (200,000 examples), regardless of variations on regularization hyper-parameters, we could make ResNet50 to overfit. Consequently, both utility and privacy are good.

In an effort to still improve Privacy, we used unsupervised domain adaptation (UDA). To that end, we use as source domains a synthetic dataset, called large Fake MNIST [34], which are similar to MNIST. Large fake MNIST has 50,000 white-on-black images for each digit, which results in 500,000 images in total. The target domain is the Defender set of QMNIST. The chosen UDA method is DSAN [32,35], which optimizes the neural network with the sum of a cross-entropy loss (classification loss) and a local MMD loss (transfer loss) [35]. We tried three variants of attacks of this UDA model. The simplest is the most effective: attack the model as if it were trained with supervised learning. Unfortunately, UDA did not yield improved performance. We attribute that to the fact that the supervised model under attack performs well on this dataset and already has a very good level of privacy.

**Table 2.** Utility and privacy of DNN ResNet50 Defender models trained on QMNIST.

| Defender Model | Utility | Privacy |
|---|---|---|
| Supervised | $1.00 \pm 0.00$ | $0.97 \pm 0.03$ |
| Unsupervised Domain Adaptation | $0.99 \pm 0.00$ | $0.94 \pm 0.03$ |

## 7. Discussion and Further Work

Although an LTU Attacker is all knowledgeable, it must make efficient use of available information to be powerful. We proposed a taxonomy based on information available or used (Figure 2). The most powerful Attackers use both the trained Defender model $\mathcal{M}_D$ and its trainer $\mathcal{T}_D$.

When the Defender trainer $\mathcal{T}_D$ is a black box, like in our first set of experiments on scikit-learn algorithms, we see clear limitations of the LTU Attacker which include the fact that it is not possible to diagnose whether the algorithm is example-based.

Unfortunately, white-box attacks cannot be conducted in a generic way, but must be tailored to the trainer (e.g., gradient descent algorithms for MLP). In contrast, black-box

methods can attack $\mathcal{T}_D$ (and $\mathcal{M}_D$) regardless of mechanism. Still, we obtain necessary conditions for privacy protection by analyzing black-box methods. Both theoretical and empirical results using black-box attackers (on a broad range of algorithms of the scikit-learn library on the QMNIST and CIFAR-10 data), indicate that Defender algorithms are vulnerable to an LTU Attacker if it overfits the training Defender data or if it is deterministic. Additionally, the degree of stochasticity of the algorithm must be sufficient to obtain a desired level of privacy.

We explored white-box attacks neural networks trained with gradient descent. In our experiments on the large QMNIST dataset (200,000 training examples), deep CNNs, such as ResNet, seem to exhibit both good utility and privacy in their "native form", according to our white-box attacker. We were pleasantly surprised with our white box attack results, but, in light of the fact that other authors found similar networks vulnerable to attack [1], we conducted the following sanity check. We performed the same supervised learning experiment by modifying 20% of the class labels (to another class label chosen randomly), in both the Defender set and Reserved set. Then we incited the neural network to overfit the Defender set. Although the training accuracy (on Defender data) was still nearly perfect, we obtained a loss of test accuracy (on Reserved data): 78%. According Theorem 2, this should result in a loss of privacy. This allowed us to verify that our white-box attacker correctly detected a loss of privacy. Indeed, we obtained a privacy of 0.55.

We are in the process of conducting comparison experiments between our white-box attacker and that of [1]. However, their method does not easily lend itself to be used with the LTU framework, because it requires training a neural network for each LTU round (i.e., on each $\mathcal{D}_A = \mathcal{D}_D - \{\texttt{membership}(d)\} \cup \mathcal{D}_R - \{\texttt{membership}(r)\}$). We are considering doing only one data split to evaluate privacy, with $\mathcal{D}_A = 50\% \, \mathcal{D}_D \, \cup \, 50\% \, \mathcal{D}_R$ and using the rest of the data for privacy evaluation. However, we can still use the pairwise testing of the LTU methodology, i.e., the evaluator queries the attacker with pairs of samples, one from the Defender data and the other from the Reserved data. In Appendix C, we show on an example that this results in an increased accuracy of the attacker.

In Appendix C, we use the same example to illustrate how we can visualize the privacy protection of individuals. Further work includes comparing this approach with [6].

Further work also includes testing LTU Attacker on a wider variety of datasets and algorithms, varying the number of training examples, training new white-box attack variants to possibly increase the power of the attacker, and testing various means of improving the robustness of algorithms against attacks by LTU Attacker. We are also in the process of designing a competition of membership inference attacks.

## 8. Conclusions

In summary, we presented an apparatus for evaluating the robustness of machine learning models (Defenders) against membership inference attack, involving an "all knowledgeable" LTU Attacker. This attacker has access to the trained model of the Defender, its learning algorithm (trainer), all the Defender data used for training, minus the label of one sample, and all the similarly distributed non-training Reserved data (used for evaluation), minus the label of one sample. The Evaluator repeats this Leave-Two-Unlabeled (LTU) procedure for many sample pairs, to compute the efficacy of the Attacker, whose charter is to predict the membership of the unlabeled samples (training or non-training data). We call such a LTU Attacker the LTU-attacker for short. The LTU framework helped us analyze privacy vulnerabilities both theoretically and experimentally.

The main conclusions of this paper are that a number of conditions are necessary for a Defender to protect privacy:

- Avoid storing examples (a weakness of example-based method, such as Nearest Neighbors);
- Ensure that $p_R = p_D$ for all $f$, following Theorem 1 ($p_R$ is the probability that discriminant function $f$ "favors" Reserved data while $p_D$ is the probability with which it favors the Defender data);

- Ensure that $e_R = e_D$, following Theorem 2 ($e_R$ is the expected value of the loss on Reserved data and $e_D$ on Defender data);
- Include some randomness in the Defender trainer algorithm, after Theorem 3.

**Author Contributions:** Conceptualization, All; Methodology, All; Software, J.P., R.M.-G., J.H. and H.S.; Validation, J.P., R.M.-G., J.H. and H.S.; Formal analysis, J.P.; Investigation, J.P., R.M.-G., J.H. and H.S.; Resources, I.G.; Data curation, J.P., R.M.-G., J.H. and H.S.; Writing—original draft preparation, All; Writing—review and editing, J.P., R.M.-G., H.S., I.G. and W.-W.T.; Visualization, J.P. and I.G.; Supervision, I.G. and W.-W.T.; Project administration, I.G. and W.-W.T.; Funding acquisition, I.G. and W.-W.T.; All authors have read and agreed to the published version of the manuscript.

**Funding:** This work is funded in part by the ANR (Agence Nationale de la Recherche, National Agency for Research) under AI chair of excellence HUMANIA, grant number ANR-19-CHIA-0022.

**Data Availability Statement:** The QMNIST dataset can be obtained at https://github.com/ facebookresearch/qmnist accessed on 4 May 2022. The CIFAR10 dataset can be obtained at https://www.cs.toronto.edu/~kriz/cifar.html accessed on 4 May 2022. Our code used to create the training/test splits of the data is available at https://github.com/JiangnanH/ppml-workshop accessed on 4 May 2022.

**Acknowledgments:** We are grateful to our colleagues Kristin Bennett and Jennifer He for stimulating discussion.

**Conflicts of Interest:** The authors declare no conflict of interest.

## Appendix A. Derivation of the Proof of Theorem 2

If the loss function $\ell(x)$ used to train the Defender model is bounded for all $x$, without loss of generality $0 \leq \ell(x) \leq 1$ (since loss functions can always be re-scaled), and if $e_R$, the expected value of the loss function on the Reserved data, is larger than $e_D$, the expected value of the loss function on the Defender data, then a lower bound on the expected accuracy of the LTU Attacker is given by the following function of the generalization error $e_R - e_D$:

$$A_{ltu} \geq \frac{1}{2} + \frac{1}{2}(e_R - e_D) \tag{A1}$$

**Proof.** If the order of the pair $(u_1, u_2)$ is random and the loss function $\ell(x)$ is bounded by $0 \leq \ell(x) \leq 1$, then the LTU Attacker could predict $u_1 \in \mathcal{D}_R$ with probability $\ell(u_1)$, by drawing $z \sim U(0, 1)$ and predicting $u_1 \in \mathcal{D}_R$ if $z < \ell(u_1)$, and $u_1 \in \mathcal{D}_D$ otherwise. This gives the desired lower bound on the expected accuracy, derived as follows:

$$A_{ltu} = \frac{1}{2} \Pr_{u_1 \sim \mathcal{D}_R}[z < \ell(u_1)] + \frac{1}{2}\left(1 - \Pr_{u_1 \sim \mathcal{D}_D}[z < \ell(u_1)]\right)$$

When $u_1$ is drawn uniformly from $\mathcal{D}_R$, then:

$$\Pr_{u_1 \sim \mathcal{D}_R}[z < \ell(u_1)] = \frac{1}{|\mathcal{D}_R|} \sum_{u_1 \in \mathcal{D}_R} Pr[z < \ell(u_1)]$$

$$= \frac{1}{|\mathcal{D}_R|} \sum_{u_1 \in \mathcal{D}_R} \ell(u_1)$$

$$= \mathbb{E}_{u_1 \sim \mathcal{D}_R}[\ell(u_1)] \tag{A2}$$

Similarly, when $u_1$ is drawn uniformly from $\mathcal{D}_D$, then:

$$\Pr_{u_1 \sim \mathcal{D}_D}[z < \ell(u_1)] = \frac{1}{|\mathcal{D}_D|} \sum_{u_1 \in \mathcal{D}_D} \Pr[z < \ell(u_1)]$$

$$= \frac{1}{|\mathcal{D}_D|} \sum_{u_1 \in \mathcal{D}_D} \ell(u_1)$$

$$= \mathop{\mathbb{E}}_{u_1 \sim \mathcal{D}_D}[\ell(u_1)] \tag{A3}$$

Substituting in these expected values gives:

$$= \frac{1}{2} \mathop{\mathbb{E}}_{u_1 \sim \mathcal{D}_R}[\ell(u_1)] + \frac{1}{2} - \frac{1}{2} \mathop{\mathbb{E}}_{u_1 \sim \mathcal{D}_D}[\ell(u_1)]$$

$$= \frac{1}{2} + \frac{e_R - e_D}{2} \tag{A4}$$

where $e_R := \mathop{\mathbb{E}}_{u_1 \sim \mathcal{D}_R}[\ell(u_1)]$ and $e_D := \mathop{\mathbb{E}}_{u_1 \sim \mathcal{D}_D}[\ell(u_1)]$

□

## Appendix B. Non-Dominance of Either Strategy in Theorem 1 or Theorem 2

Here we present two simple examples to show that neither of the strategies in Theorem 1 or Theorem 2 dominates over the other.

For example 1, assume that for $d \sim \mathcal{D}_D$ the loss function takes the two values 0 or 0.5 with equal probability, and that for $r \sim \mathcal{D}_R$ the loss function takes the two values 0.3 or 0.4 with equal probability. Then, $p_R = p_D = 1/2$ so that $p_R - p_D = 0$. However, $e_R = 0.35$ and $e_D = 0.25$, so $e_R - e_D = 0.1$.

For example 2, the joint probability mass function below can be used to compute that $(e_R - e_D) = 0.15 < (p_r - p_d) = 0.22$.

**Table A1.** Example joint PMF of bounded loss function, for $r \sim \mathcal{D}_R$ and $d \sim \mathcal{D}_D$. The attack strategy in Theorem 1 outperforms the attack strategy in Theorem 2 on these data.

|  | $l(r)$ | | | |
|---|---|---|---|---|
|  | **0** | **1/2** | **1** | **Row Sum** |
| $l(d) = 0$ | 0.24 | 0.24 | 0.12 | 0.6 |
| $l(d) = 1/2$ | 0.12 | 0.12 | 0.06 | 0.3 |
| $l(d) = 1$ | 0.04 | 0.04 | 0.02 | 0.1 |
| column sum | 0.4 | 0.4 | 0.2 | |

## Appendix C. LTU Global and Individual Privacy Scores

The following small example illustrates that the pairwise prediction accuracy in the LTU methodology is not a function of the accuracy, false positive rate, or false negative rate of predictions made on individual samples.

Let $f(x)$ be the discriminative function trained to predict the probability that a sample is in the Reserved set (i.e., predictions made using a threshold of 0.5), and for simplicity consider Defender and Reserved sets with three samples each, such that:

$$\begin{aligned} f(d_1) &= 0.1 & f(r_1) &= 0.4 \\ f(d_2) &= 0.3 & f(r_2) &= 0.7 \\ f(d_3) &= c & f(r_3) &= 0.9 \end{aligned}$$

If $c$ is either 0.6, 0.8, or 0.95, then in all three cases the overall accuracy for individual sample predictions is 2/3, and the false positive rate and false negative rate are both 1/3.

However, in the LTU methodology, the LTU Attacker would obtain 9 different pairs to predict, and for those values of $c$, its accuracy would be 8/9, 7/9, or 6/9, respectively.

We used the ML Privacy Meter python library of Shokri et al. (https://github.com/privacytrustlab/ml_privacy_meter accessed on 4 May 2022) to run their attack of AlexNet, which achieved an attack accuracy of 74.9%. Their attack model predicts the probability that each sample was in the Defender dataset. The histograms of the predictions over the Defender dataset and Reserved dataset follow:

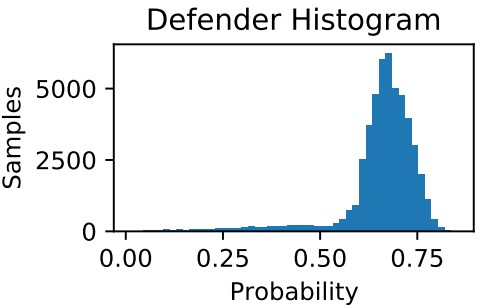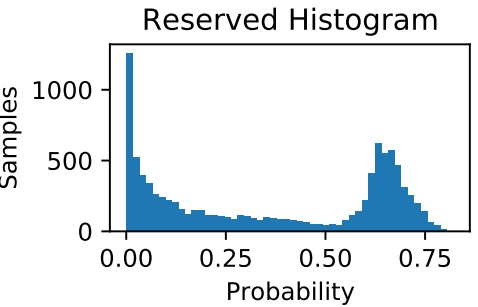

**Figure A1.** Histogram of predicted probabilities.

Using their attack model predictions in the LTU methodology, so that the attacker is always shown pairs of points for which exactly one was from the Defender dataset, increased the overall attack accuracy to 81.3%.

Furthermore, by evaluating the accuracy of the attacker on each Defender sample individually (against all Reserved samples), we computed easy to interpret individual privacy scores for each sample in the Defender dataset:

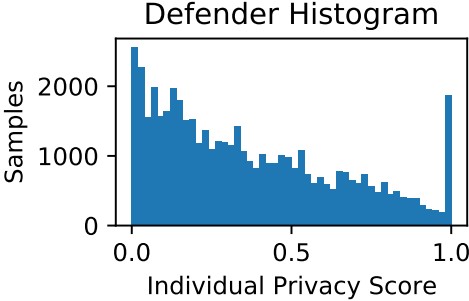

**Figure A2.** Histogram of individual privacy scores for each sample in the Defender dataset.

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
