# Peer review of "LTU Attacker for Membership Inference"

_algorithms, doi:10.3390/a15070254_

Round 1

Reviewer 1 Report

Strengths of the paper: 
1) The paper is methodologically sound and the proofs are all correct to my knowledge 
2) The LTU attacker is well explained 
3) The attack settings are well described and reasonable 
4) The experiments are sound from what I can tell 

Weakness: 
1) The paper is not very well motivated. It is unclear to me what the practical or theoretical utility of the LTU attacker is. As described in the related work section, there are papers that bound the membership inference accuracy of differentially private models, similarly there are both empirical/somewhat theoretical approaches to estimate the membership privacy of non-differentially private models. It's unclear to me where this paper fits in. Which isn't to say that the paper is of poor quality. In fact, I think it's quite methodologically sound. However, I strongly believe the authors can improve the paper by spending some time explaining why we should care about the LTU attacker/attack setting.
2) The paper is missing some relevant related work. E.g. "Nasr, Milad, et al. "Adversary instantiation: Lower bounds for differentially private machine learning." 2021 IEEE Symposium on Security and Privacy (SP). IEEE, 2021", "Thudi, Anvith, et al. "Bounding Membership Inference." arXiv preprint arXiv:2202.12232 (2022).". 
3) Table 1 has too much going on. The authors should consider simplifying it. Do we really need values bolded, rows highlighted in gray, fonts of different color? 

Author Response

Thank you for your feedback. We have tried to better motivate our framework by emphasizing that we wanted to analyze the worst case scenario (attacker has all data, all but two membership labels, and potentially white-box access to the trainer and trained model). We found that this framework allowed strong attacks using even what we refer to as naive strategies, although it did have limitations that we discussed.

Thank you for pointing us to references that we were not aware of.

Also, we have taken your recommendation to simplify Table 1.

Reviewer 2 Report

Summary:

This paper proposes a new apparatus for evaluating the robustness of machine learning models against membership inference attacks improving privacy and utility. The new apparatus is based on Leave-Two-Unlabeled (LTU) attack. The new method is experimentally evaluated on both white-box and black-box settings on QMNIST and CIFAR-10.

Strength

  1. The proposed evaluation apparatus using LTU attack seems to make sense.
  2. The experimental results show that the proposed method work on QMNIST and CIFAR-10 toy datasets.

Weakness

  1. The presentation should be drastically improved. (1) The introduction and abstract do not motivate enough why we need this new apparatus. What are the main challenges of the problem? Why did existing works fail? What’s the intuition allowing the proposed method to serve better for the task? (2) There are many unnecessarily long sentences in the paper, making the paper very hard to read. The authors should split them into simpler ones and present them in a more straightforward way.
  2. Experimental evaluation should also be improved. At least some existing membership inference attacks should be compared with the proposed attacks on the same settings in Table 1. Also, there are some membership inference specific defenses. Evaluating the new method on SOTA defenses would also be very useful in demonstrating the importance of the proposed method.
  3. Line 50 “In contrast, our LTU Attacker is not trying to address a realistic scenario”. I am not sure how to understand this sentence given the settings of the proposed method. The authors should at least clarify it either in limitations/discussions or in related works.

Author Response

Thank you for your feedback. We have tried to better motivate our framework by emphasizing that we wanted to analyze the worst case scenario (attacker has all data, all but two membership labels, and potentially white-box access to the trainer and trained model). It also enables individual privacy scores.

We also added experiments showing that our framework allowed us to conduct strong attacks against even defended models (e.g. MemGuard) using what we refer to as naive attack strategies. However, it has limitations, as we discuss.

We have edited some of the sentences to hopefully be more clear, especially the one that you highlighted.

Round 2

Reviewer 2 Report

Thank the authors for the reply and revision. Most of my concerns are addressed. The presentation and motivations are drastically improved and new interesting experimental results are provided.